# Sex-Differences in Traumatic Brain Injury in the Absence of Tau in *Drosophila*

**DOI:** 10.3390/genes12060917

**Published:** 2021-06-14

**Authors:** Ekta J. Shah, Katherine Gurdziel, Douglas M. Ruden

**Affiliations:** 1Department of Pharmacology, Wayne State University School of Medicine, Detroit, MI 48201, USA; ejshah@med.wayne.edu; 2Office of the Vice President of Research, Wayne State University, Detroit, MI 48201, USA; 3Institute of Environmental Health Sciences, Wayne State University, Detroit, MI 48201, USA; 4Department of Obstetrics and Gynecology, Wayne State University, Detroit, MI 48201, USA

**Keywords:** TBI, sex-differences, gene expression, *Tau*-KO

## Abstract

Traumatic brain injuries, a leading cause of death and disability worldwide, are caused by a severe impact to the head that impairs physiological and psychological function. In addition to severity, type and brain area affected, brain injury outcome is also influenced by the biological sex of the patient. Traumatic brain injury triggers accumulation of Tau protein and the subsequent development of Tauopathies, including Alzheimer’s disease and Chronic traumatic encephalopathy. Recent studies report differences in Tau network connections between healthy males and females, but the possible role of Tau in sex-dependent outcome to brain injury is unclear. Thus, we aimed to determine if Tau ablation would alleviate sex dependent outcomes in injured flies. We first assessed motor function and survival in *tau* knock-out flies and observed sex-differences in climbing ability, but no change in locomotor activity in either sex post-injury. Sex differences in survival time were also observed in injured tau deficient flies with a dramatically higher percent of female death within 24 h than males. Additionally, 3′mRNA-Seq studies in isolated fly brains found that tau deficient males show more gene transcript changes than females post-injury. Our results suggest that sex differences in TBI outcome and recovery are not dependent on the presence of Tau in *Drosophila*.

## 1. Introduction

Traumatic brain injury (TBI) is an insult to the brain from an external mechanical force, leading to permanent or temporary impairment of cognitive, physical and motor functions [1]. TBI causes physical injury to brain tissues that initiate molecular cascades involving unregulated neurotransmitter and ion release, inflammatory cytokines, altered gene transcription, apoptosis and free radical production [1]. TBI has long been considered a major risk factor for development of Tauopathies and other neurodegenerative diseases including Alzheimer’s disease (AD), chronic traumatic encephalopathy (CTE) and frontotemporal dementia (FTD) [2]. Although development of most Tauopathies is sporadic, brain injury can initiate misfolding and aggregation of Tau [2,3]. Under normal physiological conditions, Tau, a protein encoded in humans by *MAPT* (microtubule associated protein Tau) gene, provides cytoskeletal support and promotes microtubule stability, dynamics and axonal transport processes [3]. Tau is a natively unfolded protein that shows no tendency for aggregation by itself but post-translational modifications like phosphorylation modify the processes of Tau misfolding, oligomerization, aggregation and tangle formation [4]. Pathological Tau destabilizes the microtubule structure leading to axonal injury and damaged cytoskeleton, a common feature of brain injury [2,3].

Recent research in the area of therapeutic development for AD has shown sex-differences in the spread of abnormal Tau [5]. Positive emission tomography (PET) scans in healthy men and women and those with mild cognitive impairment (MCI), a precursor to AD, showed that healthy females not only had a stronger network of Tau protein connections but that organization was different between MCI-female and MCI-male brains [5]. In addition, female sex as a risk factor for AD has been documented in humans and transgenic models of AD and TBI [6,7]. Although TBI and AD have different etiologies, cerebrovascular dysfunction in both cases is associated with Tau and reduction or depletion of Tau have been explored as a therapeutic strategy for AD [8,9]. Interestingly, characterization of Tau deficient mice models of neurodegeneration have suggested that genetic removal of Tau is neuroprotective [8] as seen by the reduced level of axonopathy [10] and improved motor function in a Tau-deficient male mouse model of TBI [11]. It is, however, not clear whether Tau ablation would have the same effect in female models as it did in males. Thus far, there have been limited TBI studies that include both sexes in research design and none that looked at Tau ablation in both sexes in the context of TBI. Through genome-wide transcriptional changes incurred by TBI, we have previously reported vast differences in injured *w^1118^* male and female flies [7] but the exact cause of these variations remains unknown. Based on evidence from other TBI models, it is likely that Tau is a contributor of these varied outcomes in *Drosophila* and by deleting Tau, we could possibly eliminate the sex differences. Here, we used Tau-deficient (*tau*-KO) *Drosophila* to examine this hypothesis and decipher the transcriptional and behavioral changes that occur in injured flies of both sexes in the absence of Tau.

*Drosophila* have been used successfully to model TBI and flies subjected to TBI have been shown to exhibit secondary phase symptoms including innate immune response, neurodegeneration, disrupted sleep cycles and a decreased lifespan [12,13], as seen in other models of brain trauma. The fly brain is composed of a complex nervous system and expresses a Tau orthologue (dTau) that is 46% identical and 66% similar to human Tau with functions conserved from flies to humans [14]. Although we found no difference in the transcript level of Tau after TBI in *w^1118^* flies in a previous study [7], changes in the level or state of the protein could influence sex differences. Thus, we used the *Drosophila* knock-out tau mutant fly line (*tau*-KO), which lacks exons 2 to 6 including the microtubule-binding repeats and does not show any overt effects on survival, locomotion and neuronal functions in flies [15]. Since the *w^1118^* and *tau*-KO strains are on different genetic backgrounds, a direct comparison is unwarranted. In this study, we used the spring based high-impact trauma device (HIT-device) to inflict flies with TBI [16]. In the absence of Tau, we report sex-differences in motor function and survival in *tau*-KO flies post-TBI. In addition, gene expression changes in the absence of Tau within the immediate timeframe showed that *tau*-KO males have stronger gene expression response than females. Overall, our data suggests that Tau is not the only contributor of observed sex-differences in TBI outcome in *Drosophila*. 

## 2. Materials and Methods

### 2.1. Fly Stocks and Crosses

Fly stocks were stored at 25 °C at constant humidity and fed with standard sugar/yeast/agar medium. *tau*-KO (#64782) and *y^1^ w*; Mi{PT-GFSTF.0}Tau[MI03440-GFSTF.0]/TM6C, Sb^1^ Tb^1^* (Tau-EGFP) (#60199) were obtained from the Bloomington Drosophila Stock Center. All assays were performed on adult mated flies (10–14 days old).

### 2.2. Traumatic Brain Injury (TBI)

*Tau*-KO male and female flies were inflicted with single strike full body trauma using a modified high impact trauma (HIT) device with the impact arm constrained to a 45° angle [7,17,18]. No more than 25 flies were placed in a plastic vial before being confined to the bottom quarter of the vial by a stationary cotton ball. When the spring is deflected and released, the vial rapidly contacts the pad and a mechanical force is delivered to the flies as they contact the vial wall and rebound causing closed head trauma [7].

### 2.3. Confocal Imaging

About 10–15 adult Tau-EGFP flies for each sex (control and every time point post-TBI) were inflicted with TBI using the HIT-device [7,17,18]. Flies were anaesthetized with CO_2_ before brains were dissected in 1xPBS (phosphate-buffered saline) and fixed for 5 min with 4% PFA (paraformaldehyde). Fixed brains were mounted using Prolong gold antifade mounting media to visualize changes in GFP expression using a confocal microscope (Zeiss LSM 800) at the Microscopy, Imaging and Cytometry Resources Core at Wayne State University, School of Medicine. Average fluorescence intensity for all brains in each condition was calculated using ImageJ [17] in a blinded study. All data are represented as mean ± SEM. Statistical analyses (One-way ANOVA with Dunnett’s multiple comparisons test) were preformed using GraphPad Prism to compute statistical significance (*p* < 0.05) between groups.

### 2.4. Climbing Assay

For the climbing assay, 20 flies per condition (male and female; control and post TBI; 3 replicate vials each; 60 flies total) were placed in plastic vials. Flies were gently tapped to the bottom of the vials and the total number of flies that climbed above a 7 cm mark were recorded. Each vial was assessed at four time points: 10 min, 24-, 48- and 72 h after TBI. The average percent climbed across all 3 replicates is reported as mean ± SEM. Flies were maintained at 25 °C for the duration of the assay. GraphPad Prism and Two-way ANOVA with Bonferroni multiple comparisons test was used to compute significance (* *p* < 0.05) between groups. Additionally, mixed design ANOVA with TukeyHSD was also used to compute significance (# *p* < 0.05) with condition (Control or TBI) as a between-subjects factor, time (10 min, 24, 48, and 72 h) as a within-subjects factor and vial as random factor with Tukey for post hoc comparison. The mixed design ANOVA with TukeyHSD was performed in R Core Team [19]. R: A language and environment for statistical computing. R Foundation for Statistical Computing) using the “emmeans” package [7].

### 2.5. Locomotor Activity Assay

To measure locomotor activity, individual flies (24 biological replicates/condition) for control and TBI condition, were placed in tubes containing regular fly food in a *Drosophila* activity monitoring system which measures the number of times a given fly crosses an infrared beam (TriKinetics Inc., Waltham, MA, USA) [18]. The activity was assessed for 2 days. Flies were subjected to 12-h light/dark cycle with activity summarized every 30 min producing 96 timepoints of data. The number of beam breaks occurring as a result of fly movement in 30-min time-bins before the specified time-point are plotted as locomotor activity for that time-point. Flies that did not live through the recording period were not used in the calculations. 

Repeated measures ANOVA with Fisher’s Least Significant Difference (LSD) and Bonferroni for multiple comparisons test was used to compute statistical significance (*p* < 0.05) between control and TBI groups using SPSS [7].

### 2.6. Longevity Assay

Male and female flies were collected as they eclose and aged up to 10–14 days old in separate vials. After 2 weeks, flies were divided into control and TBI groups and transferred to fresh vials. Approximately 25–30 flies were placed in each vial and more than 140 flies were used for each condition (male and female; control and TBI). Flies were transferred to new vials every 2–3 days and the number of dead flies were counted each day until all flies died. Kaplan–Meier (Log-rank) test was used to compute significance (*p* < 0.05) between control and TBI conditions using SPSS. Area under the survival curve was assessed using GraphPad Prism. 

### 2.7. RNA Isolation

Total RNA was extracted from *tau*-KO single fly brains using QIAzol^®^ lysis reagent and Direct-zol™ RNA MicroPrep kit (Zymo Research, Irvine, CA, USA) following manufacturer’s instructions.

### 2.8. 3′mRNA Expression Analysis

Expression analysis was conducted in collaboration with the Wayne State University Genome Sciences Core. Three biological replicates were used for each condition (*n* = 3 for each condition: male and female at control, 1, 2 and 4 h post-TBI) and gene expression analysis was performed as described in [7,20]. 

QuantSeq 3′ mRNA-Seq Library Prep Kit FWD for Illumina (Lexogen, Greenland, NH, USA) was used to generate libraries of sequences close to the 3′ end of polyadenylated RNA from 15 ng of total RNA isolated from single fly brain in 5 µL of nuclease-free water following the low-input protocol. Library aliquots were assessed for overall quality using the ScreenTape for the Agilent 2200 TapeStation and quantified using Qubit™ 1X dsDNA HS Assay kit (Invitrogen, Waltham, MA, USA). Barcoded libraries were normalized to 2 nM before sequencing at 300 pM on one lane of a NovaSeq 6000 SP flow cell. After de-multiplexing with Illumina’s CASAVA 1.8.2 software, the 50 bp reads were aligned to the *Drosophila* genome (Build dm3) with STAR_2.4 [21] and tabulated for each gene region [22]. Differential gene expression analysis was used to compare transcriptome changes between conditions using edgeR v.3.22.3 [23] and transcripts were defined as significantly differentially expressed at absolute log2 fold change (|log2 FC|) > 1 with a false discovery rate (FDR) < 0.05. Significant gene expression changes were submitted for gene ontology analyses using RDAVID [24] for the following categories: GOTERM_BP_ALL, GOTERM_MF_ALL, UP_KEYWORDS, GOTERM_BP_DIRECT and GOTERM_MF_DIRECT. 

### 2.9. Heatmaps

Heatmaps were generated using Java Treeview [25]. Counts representing the number of reads mapped to each gene were obtained using HTSeq [22] from STAR alignments [21] before normalization. To normalize, a scaling factor was determined by dividing the uniquely mapped reads for each sample by the sample with the highest uniquely mapped number of reads. The scaling factor was multiplied to each gene count for the sample. The log2 of the normalized averaged counts for all 3 replicates is represented for each condition on the orange scale (0–10). The log2 fold change, represented on yellow-blue scale (0–6), for each gene is obtained from differential expression analysis across all 3 replicates [23]. Genes significant (|log2 FC| > 1, *p*-value < 0.05) in at least 1 time point are indicated in black text.

### 2.10. Polymerase Chain Reaction (PCR) for Genotyping

Isolated RNA was converted to cDNA using the iScript cDNA synthesis kit (Bio-Rad Laboratories, Inc., Hercules, CA, USA) and RT-PCR was performed on samples using the AmpliTaq Gold^®^ Fast PCR Master Mix Protocol (Thermo Fisher Scientific, Waltham, MA, USA). The resulting product was analyzed by gel electrophoresis and densitometry. Primer pairs for exon 3 of dTau: 5′-GTCGATGTGGGCGTTTTTAC-3′ and 5′-GCTCTGGGGTCTTGAGGAG-3′.

## 3. Results

### 3.1. Drosophila Tau-KO Exhibit Sex Differences in Motor Function after TBI Injury

Sexual dimorphism in cognitive and behavioral outcome and recovery have been observed in TBI and Tau transgenic animal models [26,27]. Several groups have explored the ablation of Tau as a therapeutic strategy in AD to prevent brain damage [28,29] but there exists only one report in a male mouse model of TBI wherein Tau reduction improved motor function after injury [10]. We wanted to assess whether varied motor function outcomes in injured male and female flies are observed in the absence of Tau. As a measure of motor function after TBI, we examined the locomotor activity and climbing ability of *tau*-KO male and female flies. 

Locomotor activity was assessed at control and TBI conditions for both *tau*-KO female and male flies using the *Drosophila* activity monitoring system (TriKinetics Inc., Waltham, MA, USA) (Figure 1A,B and Appendix A). Adult control and TBI inflicted flies were anesthetized and immediately placed in activity monitor tubes. The locomotor activity was assessed for 48 h and the activity was recorded every 30 min. The average activity in each group (n > 20 flies) at 0, 1, 2, 4 and 24 h after TBI is plotted (mean ± SEM) for *tau*-KO female and male flies (Figure 1). We observed that *tau*-KO female and male flies show varying levels of locomotor activity before and after injury. 

*Drosophila* normally exhibit negative geotactic response, i.e., when stimulated by tapping to the bottom of the vial, flies rapidly climb to the top of the vial and stay there. We used a climbing assay to assess defects in this response after injury by tapping flies to the bottom of vial and recording the number of flies that cross 70% height of the vial in 15 s. Tau-deficiency results in significant decrease in climbing ability 10 min after injury in both sexes (two-way ANOVA with Bonferroni) (Figure 1C,D). However, more than 50% of females (Figure 1C) cross the threshold 10 min after injury as compared to only ~30% males (Figure 1D). There is no significant change observed in negative geotaxis after 24 h of injury in females but males exhibit significant defects up to 72 h (Mixed design ANOVA with TukeyHSD). Overall, *tau*-KO males exhibit greater impairment in climbing ability than females. 

Although we saw no change in locomotor activity after TBI in *tau*-KO flies (Figure 1A,B, Appendix A), we saw differences in female and male climbing ability after TBI up to 72 h (Figure 1C,D and Appendix A). This demonstrate sex differences in motor function after TBI are evident in the absence of Tau. 

### 3.2. Lifespan Varies in Male and Female Tau Deficient Drosophila Post-TBI

Survivors of TBI may have a reduced lifespan compared to the general population [30]. However, the risk of TBI-mortality is found to be twice as high among males compared to females [31]. Given the complex nature of TBI pathology, we investigated the effect of Tau deficiency on long-term survival post-TBI in male and female flies by analyzing lifespan of *tau*-KO flies at control and TBI condition (Figure 2).

In *tau*-KO flies, we observed high mortality within the first 24 h after TBI, possibly due to the primary injuries from trauma [32]. However, female mortality is higher than males as more than 50% flies die within 24 h whereas 50% death is not observed in males up to 15 days post-injury. It is possible that injuries from other metabolic tissues contribute to higher mortality in females within the first 24 h. It was also observed that *tau*-KO males inflicted with TBI exhibit a 5% increase in survival time (estimated as the area under the survival curve) whereas *tau*-KO females had a 2% decrease in survival time after injury.

This data shows that Tau deficient *Drosophila* male and female flies exhibit differences in survival post-TBI. In a previous *Drosophila* TBI study it was seen that TBI results in death after injury only if the injury exceeds a certain threshold [32]. Similarly, we have also seen that death following TBI is dependent on a certain threshold of injuries, which appears to be lower for *tau*-KO females as compared to males. However, the possible causes for these sex differences in injury threshold needs further investigation. 

### 3.3. Sex-Dependent Gene Expression Changes Occur after TBI in the Absence of Tau

An integrated analysis performed using published RNA-sequencing studies in male rat TBI models showed an upregulation of *Mapt* gene 3 months after injury [33]. In our previous study, we saw no transcriptional change in Tau within 4 h in both sexes of injured *w^1118^* flies [7]. Sexual dimorphism is also observed in motor function of injured and uninjured *Tau*-KO female and male flies (Figure 1). However, we are unable to directly compare data for *w^1118^* and *tau*-KO because of non-isogenic backgrounds but since the *Drosophila* genome contains only one homolog of the Tau family [34], we suspect that its knock-out should affect Tau gene function of microtubule stabilization and polymerization. We hypothesize that if Tau is the primary contributor of sex differences, then its ablation will eliminate such differences in gene transcription. To address this hypothesis and detect changes in genome-wide transcription in the absence of Tau, we generated 3′mRNA-seq libraries from isolated *tau*-KO male and female brains at control, 1, 2 and 4 h after TBI. 

Differential gene expression analysis shows significant changes in transcription in *tau*-KO males and females in response to injury (Figure 3). *Tau*-KO males show more genes affected and more transcripts significantly downregulated (|logFC| > 2; *p*-value <0.05) at all time-points after injury when compared to control (Figure 3A–C). *Tau*-KO females inflicted with trauma exhibit fewer gene expression changes post-TBI (Figure 3D–F). At the cellular level, TBI is accompanied by activation of immune response, axonal damage, mitochondrial dysfunction, increase in stress response and cell death [35]. Significant genes identified from sequencing were classified for their biological functions using RDAVID [24] and several gene ontology (GO) categories were found to be enriched in *tau*-KO flies (Table 1 and Table 2 and Appendix A). Deficiency of Tau resulted in alteration of several genes associated with GO terms including ‘Mitochondrial Translation’, ‘Neurogenesis’ and ‘Immune response’ in *tau*-KO males (Table 2), all consistent with studies from other mammalian models [7]. In *tau*-KO males, most changes in GO categories were induced 1 h after injury with a gradual decrease in significant changes after 2 and 4 h. There were seven categories that overlapped at all time-points (Figure 4). GO terms ‘Cellular response to stimulus’ and ‘Cell communication’ were changed at all time-points indicating a response to stress and injury. *Tau*-KO females show very little change in transcription with enrichment in few GO categories after injury (Table 1). Unlike *tau*-KO males, there was no overlap seen between GO categories in *tau*-KO females across all time-points.

Overall, *tau*-KO females exhibit fewer gene expression changes compared to *tau*-KO males after injury, demonstrating an inherent sex-dependent disparity in gene expression not influenced by Tau. 

### 3.4. Immune Response to TBI Is Downregulated by Absence of Tau in Females

After brain injury, a robust neuroinflammatory response is essential to clear apoptotic cells, misfolded or aggregated proteins and pave the way for tissue repair and restore homeostasis but exaggerated and prolonged immune response can prove to be damaging [36]. A persistent inflammatory response is shown to induce acute and chronic alteration in Tau dynamics in rodents [37] while Tau-deficiency in a model of neuroinflammation has demonstrated neuroprotection [29]. Additionally, sex-differences in innate and adaptive immune responses have been well documented with reports indicating females generally exhibit a more enhanced immune response than males [38]. In a male mouse model of TBI, inhibiting microglia after TBI reduced chronic neuroinflammation and neurodegeneration [39]. Based on these findings, we assessed the change in immune response gene transcription in *tau*-KO female and male flies (Figure 5).

*Drosophila* lack an adaptive immune system and rely solely on innate immunity consisting primarily of the Toll, Immunodeficiency (Imd) and Janus Kinase protein and the Signal Transducer and Activator of Transcription (JAK-STAT) pathways [40]. We looked at changes in transcript levels of genes involved in the Toll pathway which specifically controls the transcription of the anti-fungal peptide *Drosomycin* (*Drs*) and the Imd pathway which activates the transcription of the anti-bacterial peptides *Diptericin* (*Dpt*), *Cecropin* [41] and *Attacin* (*Att*) [42] after injury in both sexes of *tau*-KO flies. In *tau*-KO females (Figure 5A), we see a decrease in immune response post-TBI as only *DptB* and *pll* (*pelle*) are significantly changed in response to injury. Conversely, *tau*-KO males (Figure 5B) exhibit a greater change in transcription of immune response genes with several transcripts significantly downregulated post-TBI. There is a significant decrease in transcription in antibacterial and antifungal effector proteins including *CecA1*, *CecA2*, *CecB*, *DptB* and *AttC* in *tau*-KO males. Although not significantly induced after injury, *Drs* is consistently expressed at high levels in both sexes. 

These data suggest that male and female flies exhibit differences in immune response post-injury in the absence of Tau. It is, however, not known how Tau depletion suppresses immune response activation after injury. 

### 3.5. Tau-Deficiency Alters Mitochondrial Transcript Levels Differently between Sexes

One of the prominent concerns after brain injury involves secondary damage associated with mitochondrial dysfunction which leads to increased production of reactive oxygen species (ROS) coupled with subsequent apoptosis and decreased cellular energy production [43]. Since mitochondria are one of the many organelles transported in anterograde and retrograde pattern along the microtubules that are stabilized by Tau, it is likely that absence of Tau culminates in alterations to mitochondrial dynamics. Interestingly, Tau deletion in young mice enhanced brain function by improving mitochondrial health [44]. Similar to Tau, sex-differences in mitochondrial function have also been studied as a major contributor to TBI outcome [45]. To understand if Tau ablation in TBI inflicted flies could limit aberrant mitochondrial gene transcription, we assessed transcriptional changes in mitochondrial genes in *tau*-KO *Drosophila* (Figure 6).

Overall, Tau ablation downregulates expression of mitochondria related genes in females (Figure 6A). *Tau*-KO males (Figure 6B) show few gene expression changes post-TBI with an initial significant downregulation of *Surf1*(*Surfeit 1*), involved in the assembly of the mitochondrial respiratory chain complex Cytochrome Oxidase at 1 h after injury and no change at later time-points. This could be indicative of an initial alteration in mitochondrial dynamics in response to injury in males in the absence of Tau. *Yip2* (*yippee interacting protein 2*), involved in mitochondrial β-oxidation is significantly downregulated in *tau*-KO males. This downregulation in energy genes may reflect an adaptive cellular response to functional and structural impairment caused by damage to neurons in response to TBI. *Vimar* (*visceral mesodermal armadillo-repeats*), a regulator of mitochondrial fission through *Miro* (*Mitochondrial Rho*), is also significantly downregulated in *tau*-KO females and males. Additionally, we have observed a significant decrease in transcripts for *aralar1*, which shuttles metabolites, nucleotides and cofactors across the inner mitochondrial membrane, and genes involved in mitochondrial biogenesis like *mRpL55* (*mitochondrial ribosomal protein L55*) and *mRpL43* (*mitochondrial ribosomal protein L43*) after injury in males. Under circumstances of decreased fuel supply or increased energy demand, such as injury or neuronal damage, cells have to adjust their metabolic processes to maintain the availability of cellular energy (as ATP) to meet the energetic needs of tissues [46]. We speculate that downregulation of these genes could indicate absence of mitochondrial impairment in response to injury in the absence of Tau.

These data show that Tau deficient males and females exhibit differences in mitochondrial gene transcription with males exhibiting more significant downregulation in expression of genes involved with solute transport and mitochondrial biogenesis. It is possible that Tau deficiency is protective against mitochondrial dysfunction, but it does not appear to be the source of sex dependent response to TBI.

In summary, our results report sex differences in TBI outcome in the *Tau*-KO flies. The exact mechanisms for varied outcomes in male and female flies need further investigation but the differences in fly brain architecture could possibly mediate these sex differences post-injury. 

## 4. Discussion

Here, we evaluated sex differences in TBI response in *Drosophila* inflicted with full body trauma in the absence of Tau. In addition to age, severity and location of impact, biological sex has been shown to influence variation in TBI outcome [7,32]. We have previously identified divergent sex dependent TBI response in transcription and motor function in *w^1118^ Drosophila* in the presence of Tau [7]. We also assessed the presence of sex differences in the presence of Tau protein using the Minos transposon-based MiMIC (Minos Mediated Integration Cassette) fly strain *y^1^ w*; Mi{PT-GFSTF.0}Tau[MI03440-GFSTF.0]/TM6C, Sb^1^ Tb^1^* that results in expression of Tau tagged with EGFP-FlAsH-StrepII-TEV-3xFlag (Tau-EGFP) [47] (Appendix A). In females, we observed a significant increase in EGFP tagged Tau expression after TBI starting at 1 h up to 24 h whereas a significant increase was observed only at 1-h post-injury in males, showing variations in EGFP-tagged Tau protein expression in response to TBI between male and female flies. After having established the presence of sex dependent TBI outcomes in flies, we questioned whether Tau is a contributing factor to these differences. We hypothesize that if the TBI outcomes in flies are influenced by the presence of Tau, then ablation of Tau could eliminate varied responses in injured flies. To address this hypothesis, we first assessed change in motor function and survival post-TBI in *Tau*-KO flies of both sexes. We also assessed gene transcription changes in isolated *tau*-KO male and female fly brains at control, 1, 2 and 4-h post-TBI to capture differences within the immediate time frame in the absence of Tau. Throughout our assessments, we observed differences in male and female response in the *Tau*-KO flies post-TBI, indicating sex dependent outcome after injury is not primarily dependent on Tau. Presence and deletion of Tau in flies was verified by PCR using primer pairs for exon 3 of dTau. At the protein level, we observed a clear signal for Tau-EGFP but no signal was detected for *Tau*-KO (Appendix A). However, a direct data comparison between *w^1118^*, Tau-EGFP and *tau*-KO flies is unwarranted because of different genetic backgrounds.

In general, TBI symptoms are thought to resolve within 3 months post-injury but a scoping review from 45 studies indicates that approximately half of individuals with a single mild TBI demonstrate long-term impairment due to pathophysiological changes in the brain [48]. Included amongst these changes is the presence of neuroinflammation, cognitive impairment, learning and memory deficits, mitochondrial damage and progressive Tau pathology observed years after injury [2]. Following TBI, a robust neuroinflammatory response develops, which is characterized by release of inflammatory mediators, cytokines, chemokines and migration of macrophages and lymphocytes [49]. Existing evidence suggests that acute and chronic activation of immune system alters Tau dynamics, but the exact mechanisms are yet to be fully elucidated [36]. In addition, disruption of mitochondrial function is also shown to be an effect of Tau pathology [50]. Thus, alteration in Tau dynamics appears to be paramount in relation to its contribution to neurodegeneration [51]. Similar to Tau, sexual dimorphism has been previously reported in mitochondrial metabolism [52] and immune function [2]. Based on these findings, we questioned whether sex differences observed in TBI response [7] could solely be dependent on similar brain-wide differences in endogenous *Drosophila* Tau and if Tau indeed was the source of sex dependent outcome then Tau deficiency will alleviate differences between sexes. Since Tau depletion in flies had no overt negative or positive effects [15], we utilized *tau*-KO flies to explore the existence of sex differences in TBI in the absence of Tau.

Impaired movement, balance and coordination are some of the initial symptoms experienced by TBI survivors [53] so we looked at motor abilities and lifespan of *tau*-KO flies after TBI. Similar to humans, flies inflicted with trauma using the HIT-device exhibit incapacitation and ataxia visible immediately after injury [32]. Here, we assessed locomotor activity and climbing ability as measures of motor function in flies (Figure 1 and Appendix A). There is a fundamental difference between these assays in that locomotor activity is independent of a mechanical stimulus whereas climbing assay relies on the inherent property of flies to move against gravity when stimulated. Surprisingly, we observed no change in locomotor activity in either sex post-TBI in the absence of Tau but there was a clear difference in the overall activity over 24 h between sexes with females exhibiting higher activity than males. In the negative geotaxis assay, *tau*-KO females exhibit greater climbing ability and recovery than males post-injury. As we previously observed differences in motor function post-TBI in male and female flies in the presence of Tau [7] and such differences are also observed here in the absence of Tau, it is possible that Tau may not be influencing these types of movements in flies, but additional analysis is needed. Survival analysis shows that Tau deficient females exhibit higher mortality within 24 h after injury compared to males (Figure 2). In humans, TBI-related deaths are seen to be higher in males than females [31]. Interestingly, we observed no significant difference in overall lifespan of *tau*-KO male and female flies inflicted with TBI as compared to controls, indicating that absence of Tau has no impact on survival in TBI inflicted *Drosophila*. Here, we observed a higher mortality in injured females but longer lasting motor impairment in injured males. It is unknown whether the sex differences in death rates is the result of males incurring more severe injuries or a result of underlying physiological differences in injury response. It is possible that although fewer males die immediately after TBI, they sustain more severe trauma impairing motor function. In injured females, the effect of trauma sustained by other metabolic tissues resulting in higher mortality immediately after trauma is a possibility yet to be explored. 

In this study, we identified genome-wide transcriptional changes that occur in isolated fly brains of both sexes at control, 1, 2 and 4 h after injury with the HIT-device (Figure 3, Figure 4, Figure 5 and Figure 6) [12]. Since we previously observed gene expression changes in mitochondrial function and immune response in the fly TBI model [7], we looked at the same categories here to facilitate a trend comparison. While planning this study, we expected to see an increase in gene transcription of immune response and mitochondrial function after injury. These hypotheses were based on previous studies from rodent, human and fly TBI models that show activation of immune response and mitochondrial dysfunction after injury [53,54]. Here, we saw mostly downregulation of gene transcription in both sexes after TBI. As for immune response, we observed an upregulation of *DptB* only in females and downregulation of *Cecropins* and *Attacins* in males post-TBI (Figure 5). Although not significantly induced after injury, *Drs* and *Mtk* are consistently expressed at high levels in both sexes. This could indicate a defense response mounted due to Tau deficiency which inhibits overexpression or activation of immune system in response to injury. It is not yet clear what mechanistic advantage is offered by downregulation of immune response in *tau*-KO males. Similar to the immune system, we saw downregulation of transcription for genes like *Surf1*, *Vimar* and *aralar1* involved in mitochondrial function (Figure 6). In a *Drosophila* model, mitochondrial enlargement due to a gain-of-function *Miro* transgene was rescued by loss-of-function vimar mutant [55]. It is also likely that *vimar* and *miro* transcription is altered in response to injury to prevent abnormal fusion and fission. Abnormal mitochondrial fission can promote cell death [55] and inhibition of mitochondrial fission has also been explored as a strategy to improve cognitive function after injury [56]. Knockdown of *aralar1* in insulinoma INS-1E cells impairs glucose oxidation correlated with a reduced generation of ATP and glutamate [57] while *Surf1* deficiency is shown to affect neurogenesis and decrease COX activity in yeast and humans [58]. It would be expected that mitochondrial activity would increase after injury to generate ATP needed to meet energy demands for cellular repair. Thus, we were surprised to see no upregulation of genes associated with the mitochondrial oxidative phosphorylation system. Several studies have employed mitochondria-targeted drugs in TBI models to increase mitochondrial biogenesis with the aim to reduce brain damage after injury [59] but here we saw downregulation of such genes namely, *mRpL55* and *mRpL43*. The absence of upregulation of immune system and mitochondrial genes could indicate that Tau deficiency is neuroprotective in brain injury.

Our previous study focused on TBI outcome in *w^1118^* flies reported varied outcomes in males and females but contrary to our expectation, we did not see a change in Tau transcription [7]. Here, we aimed to explore sex-differences in the absence of Tau and based on our findings from *Tau*-KO flies, it appears that Tau is a key player in TBI outcome. Even though we observe sex-differences in response to TBI here, the exact mechanism guiding such variations in male and female responses remains unknown. Thus, future investigations comparing TBI outcome in flies harboring Tau overexpression, wild-type and KO would be valuable to decipher the role of Tau in *Drosophila* brain injury. In conclusion, we surmise that sex differences in TBI outcome persist in the absence of Tau. It should, however, be noted that males and females have a variable response to this deficiency and the possible cause underlying such differences needs further investigation. Although the presence of metabolic tissues and sex specific gene expression could influence transcription, these data serve as a starting point to decipher gene expression changes in the brain only after traumatic injury.

## Figures and Tables

**Figure 1 genes-12-00917-f001:**
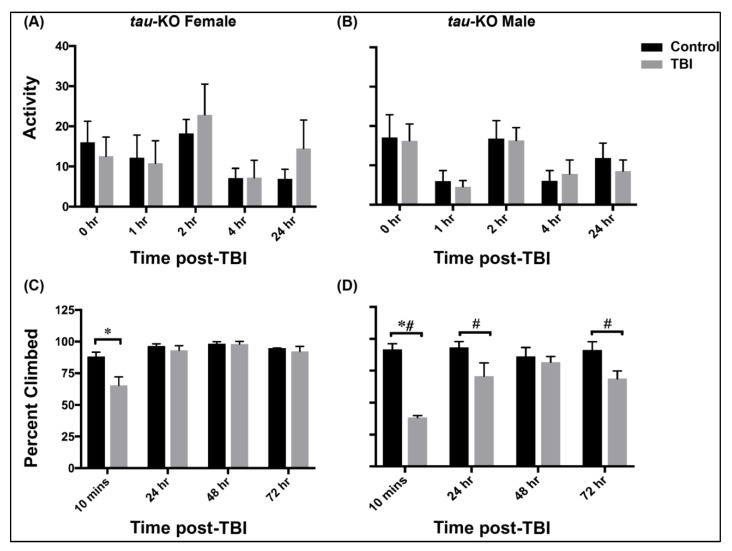
*Tau*-KO male and female flies exhibit varied outcomes to motor function assay. (**A**,**B**) Locomotor activity plots for *tau*-KO female (**A**) and male (**B**) flies after TBI. Average locomotor activity (the number of times a fly crosses the infrared beam in the 30-min before the time-point) is depicted for the first 24-h after injury as mean ± SEM (n > 20). We did not see a significant effect of TBI on locomotion (repeated measures ANOVA with LSD and Bonferroni). (**C**,**D**) Climbing ability of *tau*-KO female (**C**) and male (**D**) flies after TBI. Twenty flies per condition (male and female; control and post TBI; 3 replicate vials each) were placed in plastic vials and each vial was assessed at four time points: 10 min, 24-, 48- and 72 h after TBI. The average percent climbed across all 3 replicates is reported as mean ± SEM (each replicate consists of 20 flies; total 60 flies). Both sexes exhibit decreases in climbing ability 10 min after injury (* *p* < 0.05 with two-way ANOVA and # *p* < 0.05 with mixed design ANOVA and TukeyHSD). Males show lasting defects up to 72 h (Mixed design ANOVA with TukeyHSD).

**Figure 2 genes-12-00917-f002:**
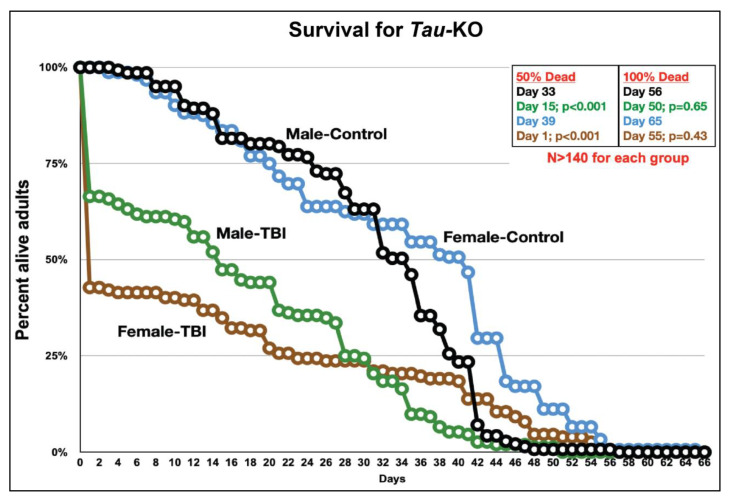
*Tau*-KO does not affect survival time in *Drosophila.* Survival curve of *tau*-KO flies at control and after TBI. Adult male and female 10–14-day old control and TBI flies were housed in separate vials. Flies were transferred into new vials every 2–3 days and the number of dead flies were counted every day till all flies died. Fifty percent death was observed in both sexes after TBI but Tau ablation did not affect the overall lifespan of TBI inflicted flies. Kaplan–Meier (Log-rank) test was used to compute statistical significance between control and TBI conditions (*p* < 0.05). The *p*-values indicated compare the differences between male-control to male-TBI (black to green) and female-control to female-TBI (blue to brown) flies at 50% and 100% death in the population.

**Figure 3 genes-12-00917-f003:**
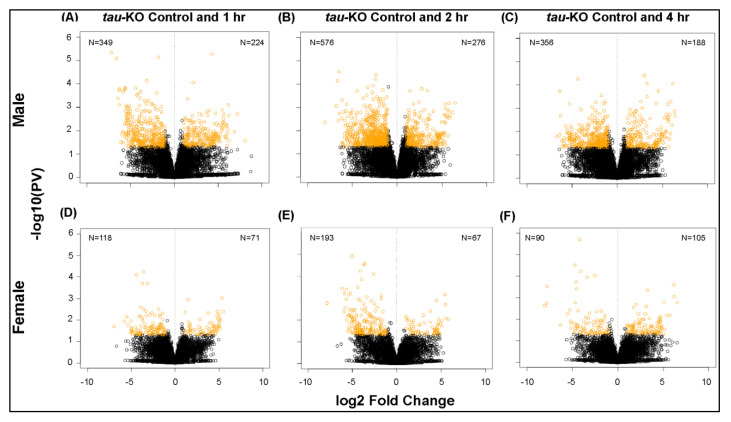
Transcriptional changes after TBI in *tau*-KO male and female flies. Volcano plots depicting log2 fold change and –log10 (PV: *p*-value) of differentially expressed genes at 1, 2 and 4 h after injury compared to control are indicated for males (**A**–**C**) and females (**D**–**F**) (*n* = 3 for each condition: male and female at control, 1, 2 and 4 h post-TBI).The number of significantly upregulated and downregulated gene changes are indicated in yellow in each plot (|log2FC| > 1; *p*-value < 0.05). *Tau*-KO males show more transcriptional upregulation of genes in response to injury than females.

**Figure 4 genes-12-00917-f004:**
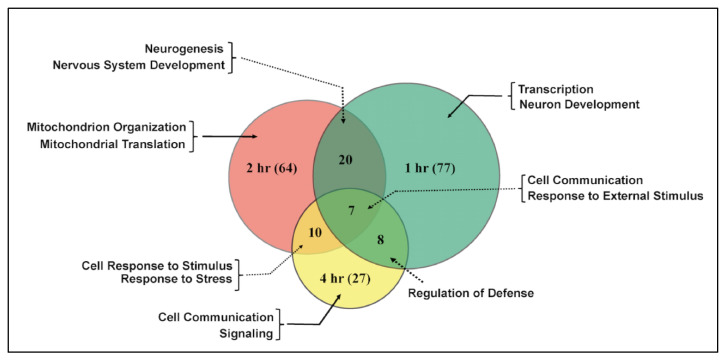
Shared gene ontology (GO) terms in *tau*-KO males across time-points after injury. Significant genes identified from sequencing were classified for their biological functions using RDAVID. The Venn diagram shows significantly changed GO terms for *tau*-KO males at 1, 2 and 4 h after injury as well as overlap between time-points. The number of GO terms differentially regulated at each time-point is indicated in parenthesis.

**Figure 5 genes-12-00917-f005:**
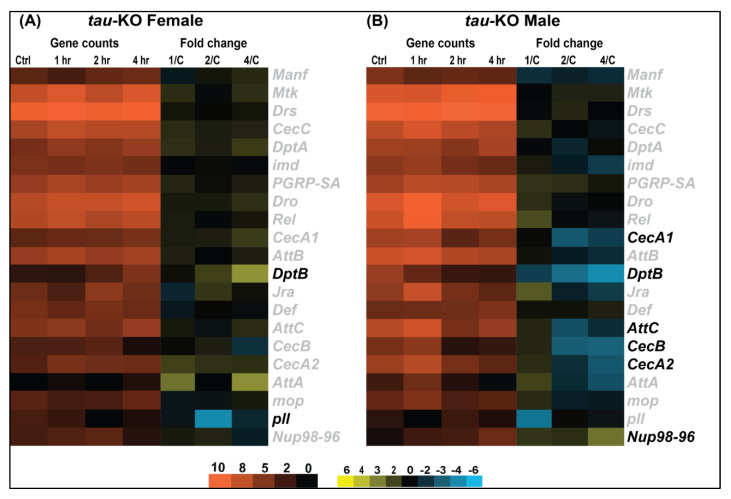
Immune response gene expression appears to be downregulated in *tau*-KO flies. Heatmaps depicting immune response gene expression changes in *tau*-KO females (**A**) and males (**B**) at control, 1, 2 and 4 h after injury. Orange scale represents the average of normalized counts for 3 replicates in each group indicated above (*n* = 3 for each condition: male and female at control, 1, 2, and 4 h post-TBI). Yellow-blue scale shows fold change for each gene at 1, 2- and 4-h post-injury compared to control. All genes indicated in black font are significant (|log2FC| > 1, *p*-value < 0.05). 1/C: Fold change at 1 h compared to control; 2/C: Fold change at 2 h compared to control and 4/C: Fold change at 4 h compared to control.

**Figure 6 genes-12-00917-f006:**
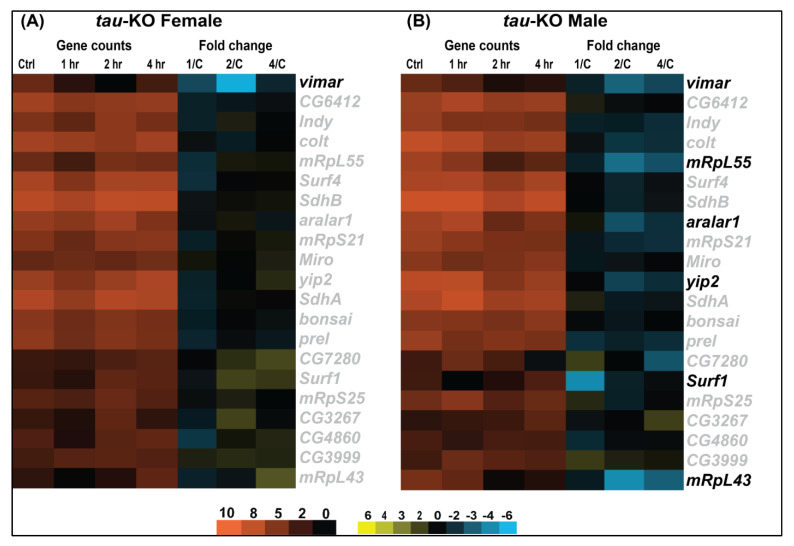
*Tau*-KO downregulates mitochondrial gene expression changes. Heatmaps depicting mitochondrial gene expression changes in *tau*-KO females (**A**) and males (**B**) at control, 1, 2 and 4 h after injury. Orange scale represents the average of normalized counts for 3 replicates in each group indicated above (*n* = 3 for each condition: male and female at control, 1, 2, and 4 h post-TBI). Yellow-blue scale shows fold change for each gene at 1, 2- and 4-h post-injury compared to control. All genes indicated in black font are significant (|log2FC| > 1, *p*-value < 0.05). 1/C: Fold change at 1 h compared to control; 2/C: Fold change at 2 h compared to control and 4/C: Fold change at 4 h compared to control.

**Table 1 genes-12-00917-t001:** Gene ontology terms significantly (FDR < 0.05) changed in response to traumatic brain injury in tau-KO female flies. GO terms were sorted based on FDR and ranked accordingly. Tables show selected GO terms changed in females after injury. GOBPID is the ID of the biological process in GO database.

**A: Selected GO Terms Differentially Regulated in *Tau*-KO Females after 1 h of Injury.**				
**Rank**	**GOBPID**	**Term**	**Fold Enrichment**	**FDR**
1	UP_KEYWORDS	Secreted	3.25	0.0134
2	GO:0007608	Sensory Perception of Smell	6.74	0.0260
**B: Selected GO Terms Differentially Regulated in *Tau*-KO Females after 2 h of Injury.**				
**Rank**	**GOBPID**	**Term**	**Fold Enrichment**	**FDR**
1	GO:0090304	Nucleic Acid Metabolic Process	1.60	<0.01
2	GO:0016070	RNA Metabolic Process	1.62	<0.01
3	GO:0046483	Heterocycle Metabolic Process	1.49	<0.01
4	GO:0006139	Nucleobase-Containing Compound Metabolic Process	1.49	<0.01
5	GO:0006725	Cellular Aromatic Compound Metabolic Process	1.45	0.0124
6	GO:1901360	Organic Cyclic Compound Metabolic Process	1.45	0.0125
7	GO:0071310	Cellular Response to Organic Substance	2.88	0.0311
**C: Selected GO Terms Differentially Regulated in *Tau*-KO Females after 4 h of Injury.**				
**Rank**	**GOBPID**	**Term**	**Fold Enrichment**	**FDR**
1	UP_KEYWORDS	Secreted	4.43	<0.01
2	GO:0007608	Sensory Perception of Smell	7.99	<0.01
3	GO:0019236	Response to Pheromone	20.83	0.0114
4	GO:0070647	Protein Modification by Small Protein Conjugation or Removal	3.02	0.0190
5	GO:0007606	Sensory Perception of Chemical Stimulus	3.32	0.0277
6	GO:0005549	Odorant Binding	4.84	0.0371
7	GO:0007608	Sensory Perception of Smell	5.12	0.0389
8	GO:0005549	Odorant Binding	4.84	0.0419

**Table 2 genes-12-00917-t002:** Gene ontology terms significantly (FDR < 0.05) changed in response to traumatic brain injury in *tau*-KO male flies. GO terms were sorted based on FDR and ranked accordingly. Tables show selected GO terms changed in males after injury. GOBPID is the ID of the biological process in GO database.

**A: Selected GO Terms Differentially Regulated in *Tau*-KO Males after 1 h of Injury.**				
**Rank**	**GOBPID**	**Term**	**Fold Enrichment**	**FDR**
1	GO:0031347	Regulation of Defense Response	3.90	<0.01
2	GO:0007154	Cell Communication	1.51	<0.01
3	GO:0007166	Cell Surface Receptor Signaling Pathway	1.97	<0.01
4	GO:0023052	Signaling	1.50	<0.01
5	GO:0009605	Response to External Stimulus	1.77	<0.01
6	GO:0044700	Single Organism Signaling	1.49	<0.01
7	GO:0050896	Response to Stimulus	1.34	<0.01
8	GO:0007165	Signal Transduction	1.56	<0.01
9	GO:0065007	Biological Regulation	1.26	<0.01
10	GO:0045088	Regulation of Innate Immune Response	4.00	<0.01
12	GO:0002682	Regulation of Immune System Process	2.77	<0.01
15	GO:0007399	Nervous System Development	1.45	<0.01
16	GO:0030182	Neuron Differentiation	1.69	<0.01
22	GO:0040011	Locomotion	1.80	<0.01
24	GO:0022008	Neurogenesis	1.48	<0.01
25	GO:0048699	Generation of Neurons	1.63	<0.01
31	GO:0050789	Regulation of Biological Process	1.24	<0.01
34	UP_KEYWORDS	Transcription	1.91	<0.01
46	GO:0010646	Regulation of Cell Communication	1.60	0.0163
47	GO:0023051	Regulation of Signaling	1.59	0.0177
55	UP_KEYWORDS	Monooxygenase	2.99	0.0269
63	GO:0010648	Negative Regulation of Cell Communication	2.05	0.0351
64	GO:0023057	Negative Regulation of Signaling	2.05	0.0351
72	GO:0045087	Innate Immune Response	2.21	0.0415
73	UP_KEYWORDS	Microsome	3.25	0.0426
**B: Selected GO Terms Differentially Regulated in *Tau*-KO Males after 2 h of Injury.**				
**Rank**	**GOBPID**	**Term**	**Fold Enrichment**	**FDR**
1	GO:0044763	Single-Organism Cellular Process	1.16	<0.01
2	GO:0007154	Cell Communication	1.41	<0.01
3	GO:0023052	Signaling	1.41	<0.01
4	GO:0006810	Transport	1.43	<0.01
5	GO:0051234	Establishment of Localization	1.39	<0.01
6	GO:0050789	Regulation of Biological Process	1.23	<0.01
7	GO:0007165	Signal Transduction	1.42	<0.01
8	GO:0006914	Autophagy	2.82	<0.01
9	GO:0022008	Neurogenesis	1.43	<0.01
10	GO:0010506	Regulation of Autophagy	3.31	<0.01
20	GO:0007399	Nervous System Development	1.31	<0.01
23	GO:0006950	Response to Stress	1.38	<0.01
28	GO:0007005	Mitochondrion Organization	1.89	0.0116
29	GO:0009966	Regulation of Signal Transduction	1.51	0.0130
35	GO:0032543	Mitochondrial Translation	2.68	0.0146
38	GO:0008219	Cell Death	1.71	0.0168
44	GO:0022008	Neurogenesis	1.56	0.0234
51	GO:0030182	Neuron Differentiation	1.42	0.0325
54	GO:0065007	Biological Regulation	1.25	0.0000
55	GO:0044262	Cellular Carbohydrate Metabolic Process	1.89	0.0391
56	GO:0006464	Cellular Protein Modification Process	1.31	0.0407
57	GO:0036211	Protein Modification Process	1.31	0.0407
59	GO:0042325	Regulation of Phosphorylation	1.91	0.0426
60	GO:0012501	Programmed Cell Death	1.68	0.0439
61	GO:0043207	Response to External Biotic Stimulus	1.69	0.0443
62	GO:0051707	Response to Other Organism	1.69	0.0443
63	GO:0030154	Cell Differentiation	1.20	0.0492
**C: Selected GO Terms Differentially Regulated in *Tau*-KO Males after 4 Hour of Injury.**				
**Rank**	**GOBPID**	**Term**	**Fold Enrichment**	**FDR**
1	GO:0043903	Regulation of Symbiosis, Encompassing Mutualism Through Parasitism	7.03	<0.01
2	UP_KEYWORDS	Amidation	7.59	<0.01
3	GO:0006950	Response to Stress	1.58	<0.01
4	GO:0007154	Cell Communication	1.38	<0.01
5	GO:0045926	Negative Regulation of Growth	3.38	<0.01
6	GO:0050792	Regulation of Viral Process	7.44	<0.01
7	GO:0050896	Response to Stimulus	1.27	<0.01
8	UP_KEYWORDS	Endoplasmic Reticulum	2.60	<0.01
9	GO:0050688	Regulation of Defense Response to Virus	8.85	<0.01
10	GO:0051093	Negative Regulation of Developmental Process	2.64	<0.01
11	GO:0044403	Symbiosis, Encompassing Mutualism Through Parasitism	4.28	<0.01
12	GO:0044419	Interspecies Interaction between Organisms	4.28	<0.01
13	UP_KEYWORDS	Aminoacyl-Trna Synthetase	6.20	<0.01
14	GO:0051241	Negative Regulation of Multicellular Organismal Process	2.56	0.0119
15	GO:0006952	Defense Response	1.94	0.0140
16	GO:0042594	Response to Starvation	2.58	0.0160
17	GO:0048640	Negative Regulation of Developmental Growth	3.56	0.0169
18	UP_KEYWORDS	Thiol Protease	5.47	0.0189
19	GO:0045886	Negative Regulation of Synaptic Growth At Neuromuscular Junction	4.72	0.0208
20	GO:0005184	Neuropeptide Hormone Activity	5.37	0.0224
21	GO:0045886	Negative Regulation of Synaptic Growth At Neuromuscular Junction	4.72	0.0233
22	GO:0051964	Negative Regulation of Synapse Assembly	4.72	0.0233
23	GO:1904397	Negative Regulation of Neuromuscular Junction Development	4.72	0.0233
24	GO:0023052	Signaling	1.33	0.0235
25	GO:0005184	Neuropeptide Hormone Activity	5.21	0.0285
26	GO:0044763	Single-Organism Cellular Process	1.13	0.0342
27	UP_KEYWORDS	Secreted	2.04	0.0351

## Data Availability

Gene expression data are available in the GEO database under accession number GSE148939. Appendix A contains *Tau*-KO Gene Ontology tables for male and female.

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
