# Peer review of "Sex-Differences in Traumatic Brain Injury in the Absence of Tau in Drosophila"

_genes, 2021, doi:10.3390/genes12060917_

Round 1

Reviewer 1 Report

The authors have done a fine job of delineating sex-differences using a model of TBI in Drosophila with tau-KO. This is an interesting review article.

I have a number of comments:

  1. On the whole, there is too much repetitive information, whether it is published or observed by the author, and is repeatedly narrated. Many of the results just state that there is a gender difference but not that it is male or female. How the results of gender differences are connected to the main pathological mechanism of TBI, such as neuroinflammation and mitochondrial dysfunction. The author should make appropriate inferences.
  2. Abstract. Abbreviations are not recommended.
  3. Last paragraph about Drosophila, duplicate information appears in the results.
  4. A detailed integrated description of all statistical methods for the entire experiment is recommended. Some parts of the experiment do not list how many groups there are and how many animals there are in each group.
  5. The content of the first paragraph of the discussion is almost repetitive.
  6. Line 473-4, neuroinflammation is actually a double-edged. Does the author's current TBI model cause inflammation, is it severe inflammation or minor? Is there animal information of RNA or protein for reference?

Author Response

  1. On the whole, there is too much repetitive information, whether it is published or observed by the author, and is repeatedly narrated. Many of the results just state that there is a gender difference but not that it is male or female. How the results of gender differences are connected to the main pathological mechanism of TBI, such as neuroinflammation and mitochondrial dysfunction. The author should make appropriate inferences.

In this study, we wanted to specifically explore whether tau-KO flies exhibit sex differences and while we see such variations in injured male and female flies, the exact mechanism behind these variations remains unknown. We have now included a statement on line 461-462 stating “Even though we observe sex-differences in response to TBI here, the exact mechanism guiding such variations in male and female responses remains unknown”.

  1. Abstract. Abbreviations are not recommended.

We have removed all abbreviations from the abstract.

  1. Last paragraph about Drosophila, duplicate information appears in the results.

We read through the manuscript for content and structure and made minor edits in the text. We hope it addresses the reviewer’s concern.

  1. A detailed integrated description of all statistical methods for the entire experiment is recommended. Some parts of the experiment do not list how many groups there are and how many animals there are in each group.

We have now included details for the number of animals in each group in figure legends.

  1. The content of the first paragraph of the discussion is almost repetitive.

We read through the manuscript for content and structure and made minor edits in the text. We hope it addresses the reviewer’s concern.

  1. Line 473-4, neuroinflammation is actually a double-edged. Does the author's current TBI model cause inflammation, is it severe inflammation or minor? Is there animal information of RNA or protein for reference? We see an increase in inflammatory gene expression but don’t know for protein

Based on our work and other published literature, it is observed that brain injury triggers an increase in inflammatory gene expression in flies but further studies are required to determine the RNA or protein expression for these inflammatory markers.

Reviewer 2 Report

The authors investigated the sex-differences in TBI in drosophila with absence of Tau.  Although the paper is well organized, there are still a few inconsistencies in the paper making it confusing. 

In section 2.4, Bonferroni method was used for multiple test correction in two-way ANOVA. However, it was not used in mixed designed ANOVA according to the manuscript. Multiple test correction is also necessary in mixed designed ANOVA.

In gene expression analysis, is is confusing to use both FDR < 0.05 and p-value < 0.05 as criteria to select significant gene. There are thousands of genes in the analysis, it is better to use FDR in figure 3 instead of p-value.

Figure 1. "Both sexes exhibit differences in overall activity". What does this mean as I did no see statistical difference between control and TBI?

Figure 2. "Tau-KO does not affect survival time in Drosophila. " We can get this conclusion when comparing Tau-KO to WT. There is no WT in this figure. There are 4 p-values in this figure. I am not sure. I think they are p-values comparing male-control to male-TBI, comparing female control to female TBI, comparing male-control to female-control, and comparing male-TBI to female-TBI. If I am right, it is misleading in the figure legend. Please give more details in the caption. 

The authors investigated immune response and changes of mitochondrial genes to TBI. How these genes were selected. There are much more genes related to immune response than the genes in the heat map. Figure 5. "Immune response gene expression is downregulated in tau-KO flies." DptB and Nup98-96 are up-regulated. It is arbitrary to make such a conclusion. 

Author Response

  1. In section 2.4, Bonferroni method was used for multiple test correction in two-way ANOVA. However, it was not used in mixed designed ANOVA according to the manuscript. Multiple test correction is also necessary in mixed designed ANOVA.

We apologize for the oversight. We did perform TukeyHSD as our multiple test correction and have now included this information in the text.

  1. In gene expression analysis, it is confusing to use both FDR < 0.05 and p-value < 0.05 as criteria to select significant gene. There are thousands of genes in the analysis, it is better to use FDR in figure 3 instead of p-value.

We agree that use of FDR has better power over the use of p-value. However, for our previous work based on same methodology, we have been instructed to use both FDR and p-value for gene expression analysis, hence we continue to do so here.

  1. Figure 1. "Both sexes exhibit differences in overall activity". What does this mean as I did no see statistical difference between control and TBI?

We have now removed the statement "Both sexes exhibit differences in overall activity" from Figure 1 to avoid confusion.

  1. Figure 2. "Tau-KO does not affect survival time in Drosophila. " We can get this conclusion when comparing Tau-KO to WT. There is no WT in this figure. There are 4 p-values in this figure. I am not sure. I think they are p-values comparing male-control to male-TBI, comparing female control to female TBI, comparing male-control to female-control, and comparing male-TBI to female-TBI. If I am right, it is misleading in the figure legend. Please give more details in the caption.

Further details regarding p-values are now included in the figure legend: The p-values indicated compare the differences between male-control to male-TBI (black to green) and female-control to female-TBI (blue to brown) flies at 50% and 100% death in the population.

  1. The authors investigated immune response and changes of mitochondrial genes to TBI. How these genes were selected. There are much more genes related to immune response than the genes in the heat map. Figure 5. "Immune response gene expression is downregulated in tau-KO flies." DptB and Nup98-96 are up-regulated. It is arbitrary to make such a conclusion.

We have now included the statement on line 426-428 “Since we previously observed gene expression changes in mitochondrial function and immune response in the fly TBI model (Shah et al., 2020), we looked at the same categories here to facilitate a trend comparison”.

Reviewer 3 Report

The work entitled “Sex-differences in traumatic brain injury in the absence of Tau  2 in Drosophila” is well written and well prepared. However, there are some minor issues related to this paper:
This sentence is awkward, please correct: “Although our previous publication (Shah et al., 2020)  showed no difference in the transcript level of Tau after TBI in w1118 flies, changes in the level or state of the protein could influence sex differences.”

“we did observe a difference in overall activity between sexes (Fig. 1AB, Fig. S1).”  - could you provide any statistical analysis to this statement?
Could you indicate whether differences between sexes in survival rates (Figure 2) are statistically significant?

This sentence in Results is hard to understand” Drosophila genome contains only one homolog of the Tau family (Papanikolopoulou et al., 2019), we suspect that its absence should affect gene function.” By absence, you mean KO, but what and which gene function?
I have one question related to the methodology of the TBI. Since authors mention that increased mortality maybe related to trauma inflicted to other organs than brain – is there any possibility to discriminate brain-only trauma from other organs? Discussion of this issue is needed.

Author Response

  1. This sentence is awkward, please correct: “Although our previous publication (Shah et al., 2020) showed no difference in the transcript level of Tau after TBI in w1118 flies, changes in the level or state of the protein could influence sex differences.”

We have now reworded the statement on line 72-83.

  1. “we did observe a difference in overall activity between sexes (Fig. 1AB, Fig. S1).” - could you provide any statistical analysis to this statement?

We have reworded lines 219-220 and removed the statement “we did observe a difference in overall activity between sexes (Fig. 1AB, Fig. S1)” to avoid confusion.

  1. Could you indicate whether differences between sexes in survival rates (Figure 2) are statistically significant?

Survival rate comparison between male-control and female-control were not found to be statistically significant (p = 0.13).

  1. This sentence in Results is hard to understand” Drosophila genome contains only one homolog of the Tau family (Papanikolopoulou et al., 2019), we suspect that its absence should affect gene function.” By absence, you mean KO, but what and which gene function?

We have now reworded the statement on line 250-251 for better clarity.

  1. I have one question related to the methodology of the TBI. Since authors mention that increased mortality maybe related to trauma inflicted to other organs than brain – is there any possibility to discriminate brain-only trauma from other organs? Discussion of this issue is needed.

Previous Drosophila studies have observed intestinal and blood–brain barrier dysfunction after full body trauma (Katzenberger, 2015) and immune response changes after a head-specific trauma (Alphen 2018). However, a combined study employing an alternate trauma-inflicting device would be needed to discriminate the changes arising from brain and other organs in response to injury.

Round 2

Reviewer 1 Report

All the points raised have been addressed. 

Author Response

Thank you for your support of our work.

Reviewer 2 Report

The authors have addressed most of my questions and comments except #4. 

In figure 2, "Tau-KO does not affect survival time in Drosophila". In figure 2, the authors only compared control to TBI. We can only get the influence from TBI from this comparison.  If the authors want to know the influence of Tau-KO, they should compare survival of Tau-KO to the survival of wild type under different conditions. 

For 100% death in the population, we can still see the difference of control and TBI from the figure. It is better the authors could double check the test.

Author Response

Response to Reviewer 2 comments
In figure 2, "Tau-KO does not affect survival time in Drosophila". In figure 2, the authors only compared control to TBI. We can only get the influence from TBI from this comparison. If the authors want to know the influence of Tau-KO, they should compare survival of Tau-KO to the survival of wild type under different conditions.
For 100% death in the population, we can still see the difference of control and TBI from the figure. It is better the authors could double check the test.

Answers: Although unwarranted in this study, we agree that comparing the data for tau-KO line to a wild type will be useful to assess role of Tau. In our previous publication, we aimed to see if TBI outcome is different in male and female flies, but we were surprised to see no change in Tau transcription. Thus, in this study, we wanted to specifically explore whether tau-KO flies exhibit sex differences. Since we observed varied outcome in Tau deficient flies here, we next intend to
look at hyperphosphorylated Tau or overexpression of Tau and compare the data to wild-type and KO flies (Line 526-539).
As for the 100% death statistics, we have re-checked the survival time and found that there is no difference in the overall lifespan of Tau-KO control and TBI flies.
